# A Four-Week High-Fat Diet Induces Anxiolytic-like Behaviors through Mature BDNF in the mPFC of Mice

**DOI:** 10.3390/brainsci14040389

**Published:** 2024-04-17

**Authors:** Huixian Huang, Jia Huang, Wensi Lu, Yanjun Huang, Ran Luo, Luqman Bathalian, Ming Chen, Xuemin Wang

**Affiliations:** 1Key Laboratory of Mental Health of the Ministry of Education, Guangdong-Hong Kong-Macao Greater Bay Area Centre for Brain Science and Brain-Inspired Intelligence, Guangdong Province Key Laboratory of Psychiatric Disorders, Department of Neurobiology, School of Basic Medical Sciences, Southern Medical University, Guangzhou 510515, China; huixianhuang1073@163.com (H.H.); jiahuang1219@163.com (J.H.); xl786142581@163.com (W.L.); huangyanj123@163.com (Y.H.); 15977862285@163.com (R.L.); luqman0328@163.com (L.B.); shycm@smu.edu.cn (M.C.); 2National Demonstration Center for Experimental Education of Basic Medical Sciences, Southern Medical University, Guangzhou 510515, China

**Keywords:** high-fat diet, anxiolytic-like behaviors, mPFC, BDNF, dendritic spine

## Abstract

The effect of a high-fat diet (HFD) on mood is a widely debated topic, with the underlying mechanisms being poorly understood. This study explores the anxiolytic effects of a four-week HFD in C57BL/6 mice. Five-week-old mice were exposed to either an HFD (60% calories from fat) or standard chow diet (CD) for four weeks, followed by cannula implantation, virus infusion, behavioral tests, and biochemical assays. Results revealed that four weeks of an HFD induced anxiolytic-like behaviors and increased the protein levels of mature brain-derived neurotrophic factor (mBDNF) and phosphorylated tyrosine kinase receptor B (p-TrkB) in the medial prefrontal cortex (mPFC). Administration of a BDNF-neutralizing antibody to the mPFC reversed HFD-induced anxiolytic-like behaviors. Elevated BDNF levels were observed in both neurons and astrocytes in the mPFC of HFD mice. Additionally, these mice exhibited a higher number of dendritic spines in the mPFC, as well as upregulation of postsynaptic density protein 95 (PSD95). Furthermore, mRNA levels of the N6-methyladenosine (m6A) demethylase, fat mass and obesity-associated protein (FTO), and the hydrolase matrix metalloproteinase-9 (MMP9), also increased in the mPFC. These findings suggest that an HFD may induce FTO and MMP9, which could potentially regulate BDNF processing, contributing to anxiolytic-like behaviors. This study proposes potential molecular mechanisms that may underlie HFD-induced anxiolytic behaviors.

## 1. Introduction

Recently the potential influence of dietary patterns on mental health has been increasingly recognized. A high-fat diet (HFD) has been predominantly associated with a variety of negative health outcomes, including cognitive function impairment [1], circadian rhythm disruption [2], and even the increased risk of depression [3]. Nevertheless, the impact of an HFD on anxiety remains contentious, with studies reporting both anxiety-inducing [4,5] and anxiety-reducing effects [6]. Our previous study showed that an HFD over a four-week period elicited anxiolytic-like behaviors in adult mice [7]. Furthermore, in humans, consuming comfort foods rich in fats and carbohydrates can elevate one’s mood, imparting a sense of cheer and potentially enhancing overall well-being and performance [8]. Women experiencing high stress and greater eating exhibited a diminished cortisol response, reduced diurnal cortisol levels, and increased suppression in response to dexamethasone [9]. However, stress eating may contribute to obesity and cardiovascular diseases, and it is important to understand the underlying psychobiological mechanisms. Moreover, anxiety disorder is highly prevalent in adolescents [10]; however, the significant side effects associated with anti-anxiety medications constrain their application in teenagers. Therefore, understanding how HFD can produce anxiolytic effects in adolescents is crucial for identifying effective targets for anxiolytic treatments. In our current study, we examined the anxiolytic effects and mechanisms of a four-week HFD in C57BL/6 mice at five weeks of age, corresponding to the adolescent phase in human development.

Brain-derived neurotrophic factor (BDNF) is a crucial neuropeptide involved in various aspects of the nervous system, such as neurodevelopment and synaptic plasticity [11]. Stress exposure in rodents decreases BDNF in the hippocampus [12,13] and dorsal raphe nucleus [14]. Moreover, decreased serum levels of BDNF have been found in patients with major depressive disease. However, when it comes to general anxiety disorder (GAD), controversial results have been reported. Some studies found a significant reduction in serum BDNF levels among those diagnosed with GAD [15,16]. However, no substantial differences in serum BDNF levels related to GAD were reported in investigations conducted by Carlino et al. [17] and Lhullier et al. [18]. In contrast to these findings, Pallanti et al. identified increased serum BDNF levels in GAD patients [19]. The BDNF gene Val66Met (rs6265) single nucleotide polymorphism (SNP) is a genetic variant which has been widely researched for its association with susceptibility to anxiety in mice [20]. Frustaci et al. conducted a meta-analysis revealing that individuals with the homozygous Met allele may have a heightened risk of generalized anxiety [12]. However, this association has been debated, as further studies have yielded varying outcomes when assessing the prevalence of anxiety symptoms in carriers of the Val and Met alleles. Moreover, the Val66Met rat model, characterized by a valine-to-methionine substitution at position 66 in the rat BDNF gene, exhibited no alterations in anxiety-related behaviors [21]. Therefore, the fundamental physiology of BDNF in anxiety should be further studied.

In the present study, we investigated the impact of a four-week HFD on anxiolytic-like behaviors in mice at five weeks of age, corresponding to the adolescent phase in human development. We aimed to elucidate the relationship between HFD consumption and anxiety, with a focus on the role of BDNF in the medial prefrontal cortex (mPFC). By examining changes in BDNF levels and the expression of N6-methyladenosine (m6A)-associated genes in the mPFC, we sought to provide insights into how HFD-induced alterations in BDNF processing may contribute to anxiolytic-like behaviors. Additionally, we aimed to propose potential therapeutic targets for the treatment of anxiety disorders related to dietary patterns, particularly HFD consumption.

## 2. Materials and Methods

### 2.1. Animals

Male C57BL/6 mice at the age of four weeks were obtained from the Guangdong Medical Laboratory Animal Center in Foshan, China. The mice were kept in group housing (three to five per cage) under controlled temperature conditions (22–25 °C) with a 12 h light/dark cycle. Lighting was provided from 8:00 a.m. to 8:00 p.m., and the mice had unrestricted access to both food and water. The experimental groups were assigned randomly. The mice were acclimated to the environment for one week and then started on either an HFD or standard chow diet (CD), at the age of five weeks, for four weeks. The HFD (Research Diets, New Brunswick, NJ, USA, D12492) and CD (Research Diets, New Brunswick, NJ, USA, D12450B) contained 60% and 10% of calories from fat, respectively.

### 2.2. Behavioral Tests

All behavioral tests were conducted during the mice’s active phase. Before the tests, the mice underwent handling for a minimum of three consecutive days (5 min each day) to minimize stress during the experiments.

#### 2.2.1. Open Field Test (OFT)

The OFT was performed as previously described [22,23,24,25]. Mice were placed in the center of an open field chamber (40 cm × 40 cm × 30 cm) (Accuscan Instruments, Columbus, OH, USA) and allowed to freely explore for 30 min. Behavioral tests were recorded by a sensitive video recorder connected to a VersaMax analyzer (OMNITECH, Sioux Falls, SD, USA). We evaluated locomotor activity by analyzing the total distance traveled by the mice in 30 min. Additionally, anxiety levels were assessed by analyzing the duration of time spent in the center zone (25 cm × 25 cm) during the first 5 min.

#### 2.2.2. Light-Dark Box Test (LDT)

The LDT was performed as previously described [26,27]. A black plastic enclosure (40 cm × 20 cm × 12 cm) was positioned on one side of the open field arena (40 cm × 40 cm × 30 cm) (Accuscan Instruments, Columbus, OH, USA), partitioning the arena into two equal-sized compartments. Mice were placed in the light compartment and allowed to explore freely for 5 min. Behavioral tests were recorded by a video recorder connected to a VersaMax analyzer (OMNITECH, Sioux Falls, USA). We quantified the amount of time animals spent in the light chamber.

#### 2.2.3. Elevated Plus Maze Test (EPM)

The EPM was performed as previously described [23,28]. The EPM apparatus consisted of two open arms (30 cm × 5 cm × 0.5 cm), two enclosed arms (30 cm × 5 cm × 1.5 cm), and a center platform (5 cm × 5 cm) elevated 50 cm above the underlying surface (Yingling, Guangzhou, China). Mice were placed on the center platform of the maze and allowed to explore freely for 5 min. Behavioral tests were recorded by a video recorder with EthoVision XT 11.5 software (Noldus Information Technology, Wageningen, The Netherlands). The duration of time spent in the open arms and the number of entries into the two open arms were quantified.

### 2.3. Immunofluorescence

After intraperitoneal injections of 80 mg/kg sodium pentobarbital for deep anesthesia, mice were opened at their abdominal cavities and, after exposure of the heart, they were perfused with 0.9% saline followed by 4% paraformaldehyde (Yuanye Bio-Technology, Shanghai, China, P20497) in 0.1 M PBS, pH 7.4 (Solarbio, Beijing, China, P1003). Subsequently, the brains were removed, postfixed overnight in 4% paraformaldehyde at 4 °C, and then transferred to a 30% sucrose solution (Sigma-Aldrich, Darmstadt, Germany, V900116) in 0.1 M PBS, pH 7.4. After complete dehydration of the mouse brains, coronal sections of 40 μm were cut using a cryostat (Leica, Wetzlar, Germany, CM3000). After washing the brain sections three times with PBS, they were incubated at room temperature for 1 h with blocking solution containing 5% normal bovine serum albumin (Solarbio, Beijing, China, 9048-46-8) and 0.3% Triton X-100 (Sigma-Aldrich, Darmstadt, Germany, 93443). Subsequently, the brain sections were incubated with primary antibodies overnight at 4 °C followed by three washes with PBS. The sections were then incubated with secondary antibodies at room temperature for 1 h. The nuclei were stained with DAPI (Sigma-Aldrich, Darmstadt, Germany, D9542) at room temperature for 10 min, followed by three washes with PBS. Finally, the coverslips were mounted onto glass slides with an anti-fade solution. The brain sections were imaged using a confocal fluorescence microscope (Nikon, Tokyo, Japan).

The following primary antibodies were used: anti-BDNF (Abcam, Cambridge, UK, ab108319, 1:500), anti-NeuN (Merck Millipore, Darmstadt, Germany, ABN78, 1:500), anti-S100β (Abcam, Cambridge, UK, ab52642, 1:500), and Alexa Fluor 488/594 secondary antibodies (ZSGB-BIO, Beijing, China, ZF-0512, ZF-0511, ZF-0513, 1:500).

### 2.4. Western Blotting

Animal brain tissues were collected and homogenized in ice-cold RIPA lysis buffer (Solarbio, Beijing, China, R0010) supplemented with a proteinase inhibitor cocktail (MedChemExpress, Monmouth Junction, NJ, USA, HY-K0010) and phosphatase inhibitors (MedChemExpress, NJ, USA, HY-K0021). The homogenization process was performed on ice for 30 min, followed by centrifugation at 4 °C for 15 min at 13,000 rpm. The supernatants were carefully collected and transferred into new 1.5 mL tubes.

Total protein was quantified using a BCA assay kit (Thermo Fisher Scientific, Waltham, MA, USA, A53225). Approximately 20–30 μg of total protein was resolved by SDS/PAGE and electrotransferred to PVDF membranes (Merck Millipore, Germany, IPVJ00010). Primary antibodies were incubated with membranes at 4 °C overnight. Secondary antibodies were incubated with membranes at room temperature for 1 h. An ECL detection system (Bio-Rad, Hercules, CA, USA) was used to develop signals. The following primary antibodies were used: anti-BDNF (Abcam, Cambridge, UK, ab108319, 1:1000), TrkB rabbit mAb (Cell Signaling Technology, Danvers, MA, USA, #4603, 1:1000), p-TrkB rabbit mAb (Cell Signaling Technology, Danvers, MA, USA, #4621, 1:1000), PSD95 rabbit mAb (Cell Signaling Technology, Danvers, MA, USA, #3450, 1:1000), GAPDH monoclonal antibody (Beijing Ray Antibody Biotech, Beijing, China, RM2002, 1:10,000), β-actin monoclonal antibody (Beijing Ray Antibody Biotech, Beijing, China, RM20021, 1:10,000), Horseradish peroxidase (HRP)-conjugated goat anti-mouse and goat anti-rabbit secondary antibodies (ZSGB-BIO, Beijing, China, ZB-2301, ZB-2305, 1:5000).

### 2.5. Real-Time Quantitative PCR

After swiftly decapitating the mice, we extracted the mouse brains and then dissected the mPFC using a mouse brain mold (RWD, Shenzhen, China, 68708), based on coronal sections ranging from bregma 2.4 mm to bregma 1.4 mm as specified in the mouse brain atlas. The mPFC tissues were homogenized with TRIzol reagent (Thermo Fisher Scientific, Waltham, USA, 15596026CN), and the concentrations were quantified using a spectrophotometer (Thermo Scientific, USA, NanoDrop 2000). Total RNA was converted to cDNA using the PrimeScript RT Master Mix kit (Takara, Shiga, Japan, RR036A). Standard real-time quantitative PCR was performed using Power SYBR Green PCR Master Mix (Takara, Shiga, Japan, RR420A) on a Fast Real-Time PCR System (Applied Biosystems, Waltham, MA, USA, 7500). The gene expression analysis utilized the ΔΔCt method, with the relative mRNA levels were normalized to those of GAPDH mRNA.

### 2.6. Stereotaxic Injection of Virus

Mice were anaesthetized with sodium pentobarbital (intraperitoneally, 80 mg kg^−1^) before surgery. Next, the mice were placed in a mouse stereotaxic instrument (RWD, Shenzhen, China, 68801). A micro-injection pump (RWD, Shenzhen, China, KDS LEGATO 130) was used to deliver the virus to the target areas at a rate of 0.1 μL per minute. The virus was the dual-AAV sparse labelling of pyramidal neurons system (BrainVTA, Wuhan, China) [29,30]: rAAV-Ef1a-DIO-EYFP-WPRE-PA and pAOV-CaMKIIa-GFP-2A-cre mixture at a ratio of 1:100,000. Mice were bilaterally injected with a total of 0.2 μL of virus in their mPFC (AP: +1.80 mm, ML: ±0.4 mm, DV: −2.65 mm, relative to bregma). After each injection, the needle (Hamilton, Bonaduz, Switzerland, 87919) was left in place for 5 min and then slowly withdrawn. Subsequent experiments were performed at least three weeks after virus injection to allow sufficient time for transgene expression. Brain sections were collected for confocal imaging.

### 2.7. Cannula Implantation and BDNF-Neutralizing Antibody Administration

Mice were anesthetized with pentobarbital intraperitoneally before cannula implantation. Bilateral cannulas (RWD, Shenzhen, China, 62060) were stereotaxically aimed at the mPFC with coordinates (AP: +1.80 mm, ML: ±0.4 mm, DV: −1.65 mm, relative to bregma) and they were secured to the skulls with dental acrylic (ShangChi, Changshu, China). To prevent clogging, a catheter cap (RWD, Shenzhen, China, 62131) was placed in the cannula. Animals were allowed to recover from surgery for at least 7 days before the experiment.

For BDNF-neutralizing antibody (Merck Millipore, Germany, AB1513P) [31] or artificial cerebrospinal fluid (ACSF) infusion, a needle (GAOGE, Shanghai, China, LG01-104-5) with its tip protruding 1.0 mm beyond that of the guide was used. The infusion needles were connected to a microsyringe pump (RWD, Shenzhen, China, R462) through an acrylic tube. BDNF-neutralizing antibody or ACSF was infused at a rate of 0.1 μL per minute, 0.25 μL per side (1 μg/μL). Mice could freely explore during the infusion. The needle was left in place for an additional 5 min after infusion to facilitate diffusion and prevent reflux. Infusion was administered 24 h before each behavioral experiment.

### 2.8. Statistical Analysis

All experiments and data analyses were performed under double-blind conditions to minimize bias. Data were analyzed using SPSS 20.0 software (IBM, Armonk, NY, USA) and Graph-Pad Prism 8 (Graph-Pad, San Diego, CA, USA), with appropriate inferential methods, and are presented as the mean ± standard error of the mean (SEM). Fluorescence densities were analyzed using ImageJ 2.0.0 (National Institutes of Health, Stapleton, NY, USA). All data were analyzed by Shapiro–Wilk tests for Gaussian distribution. Two-tailed Student’s *t*-tests (parametric tests or non-parametric tests), one-way ANOVAs or two-way repeated measures ANOVAs (Bonferroni’s multiple comparisons tests) were used for statistical analyses, and differences were considered to be significant if *p* < 0.05. Significance levels are indicated as follows: * *p* < 0.05, ** *p* < 0.01.

## 3. Results

### 3.1. Mice Fed with an HFD for Four Weeks Exhibit Anxiolytic-like Behaviors

We fed male mice aged five weeks with either a CD (10% calories from fat) or an HFD (60% calories from fat) (Figure 1A). Body weights were measured weekly. After 3 weeks on the HFD, a significant increase in body weight was observed in the HFD group (time: *p* = 0.035; weight: *p* < 0.0001) (Figure 1B). Following four weeks of HFD feeding, we conducted anxiety-related behavioral tests, including the EPM, LDT, and OFT. In the EPM, the HFD group spent more time in the open arms of the maze (*p* = 0.0073) and exhibited a significantly increased number of entries into the open arms of the maze (*p* = 0.0130) compared to the CD group (Figure 1C). In the LDT, the HFD group spent more time in the light chamber (*p* = 0.0430) (Figure 1D). In the OFT, both the HFD group and CD group spent a similar duration in the center zone during the first 5 min (*p* = 0.3409) and traveled almost the same total distance during the 30 min (*p* = 0.8943) (Figure 1E), indicating that there were no significant differences in locomotor activity between the two groups.

### 3.2. mBDNF and p-TrkB Levels in the mPFC Are Increased in HFD Mice

Next, we measured the protein expression levels of BDNF in three classical brain regions known to regulate anxiety and stress: mPFC, amygdala, and hippocampus. After HFD feeding for four weeks, we observed a significant increase in mature brain-derived neurotrophic factor (mBDNF) expression (*p* = 0.0185) in the mPFC, while no significant change was found in the pro-protein of BDNF (proBDNF) levels (*p* = 0.5788) (Figure 2A). However, there were no notable differences in mBDNF (*p* = 0.5530) or proBDNF (*p* = 0.8619) expression observed in the amygdala (Figure 2B). In the hippocampus, we observed a significant decrease in proBDNF expression (*p* = 0.0210), with no significant change in mBDNF levels (*p* = 0.7791) (Figure 2C). Additionally, in the adult brain, BDNF predominantly binds to the tyrosine kinase receptor B (TrkB) receptor [32]. We examined phosphorylated tyrosine kinase receptor B (p-TrkB) and TrkB via western blotting and found an increase in p-TrkB expression (*p* = 0.0210) in the mPFC of the HFD group, while TrkB expression (*p* = 0.4325) remained unchanged (Figure 2D).

### 3.3. The Administration of a BDNF-Neutralizing Antibody into the mPFC Reverses Anxiolytic-like Behaviors Induced by the HFD

We performed cannula infusions of BDNF-neutralizing antibody or ACSF into the mPFC (Figure 3A,B). In the EPM, the HFD + ACSF group exhibited increased exploration time in the open arms, which was subsequently reversed by administration of the BDNF-neutralizing antibody (CD + ACSF vs. HFD + ACSF: *p* = 0.0367; HFD + ACSF vs. HFD + anti-BDNF: *p* = 0.0241) (Figure 3C). In the LDT, the BDNF neutralizing antibody-administered group (HFD + anti-BDNF) spent less time in the light chamber area than the HFD + ACSF group (CD + ACSF vs. HFD + ACSF: *p* = 0.0573; HFD + ACSF vs. HFD + anti-BDNF: *p* = 0.0296) (Figure 3D). Additionally, there were no significant differences among the three groups in terms of the duration spent in the center zone during the first 5 min (CD + ACSF vs. HFD + ACSF: *p* = 0.5336; HFD + ACSF vs. HFD + anti-BDNF: *p* = 0.9763) or total distance traveled during the 30 min (CD + ACSF vs. HFD + ACSF: *p* = 0.9662; HFD + ACSF vs. HFD + anti-BDNF: *p* = 0.9986) in the OFT (Figure 3E).

### 3.4. BDNF Levels Are Increased in Both Neurons and Astrocytes in the mPFC of Mice Fed an HFD

To further explore which cell types in the mPFC exhibit increased BDNF expression associated with anxiolytic-like behaviors induced by HFD, we performed immunofluorescent co-localization staining using NeuN as a neuronal cell nucleus marker and S100β as an astrocyte marker, along with BDNF staining. The immunofluorescence results indicate that after four weeks of an HFD, the levels of BDNF not only increased within the neurons (*p* = 0.0323) (Figure 4A) but also within the astrocytes (*p* = 0.0412) (Figure 4B).

### 3.5. A Four-Week HFD Promotes the Expression of PSD95 and Increases the Density of Dendritic Spines

Some studies have reported that BDNF-TrkB signaling modulates synapse formation [11]. In our study, we examined the protein expression of postsynaptic density protein 95 (PSD95), a key marker of postsynaptic integrity. We observed increased levels of PSD95 protein in the mPFC of the HFD group (*p* = 0.007) (Figure 5A).

Given that PSD95 is a major regulator of spine maturation and postsynaptic development, we next evaluated dendritic spine density. We utilized the dual-virus sparse labeling system (pAOV-CaMKIIa-GFP-2A-cre and rAAV-Ef1a-DIO-EYFP-WPRE-PA virus mixture) to specifically label pyramidal neurons in the mPFC. These viruses were injected into the mPFC of mice after one week of CD or HFD feeding. After three weeks, we examined dendritic spine density. Our results showed that the dendritic spine density in the mPFC was significantly increased in the HFD group (*p* = 0.0254) (Figure 5B).

### 3.6. FTO and MMP9 Levels in the mPFC Are Increased in HFD Mice

Next, we investigated the mechanisms underlying the HFD-induced increased expression of mBDNF in the mPFC. Previous research has demonstrated that FTO is involved in the formation of mBDNF by regulating matrix metalloproteinase-9 (MMP9) [33]. First, we examined the mRNA levels of methylases or demethylases of m6A (Table 1). Real-time quantitative PCR was employed to detect the expression of methylases—an m6A ‘writer’, methyltransferase-like 3 (Mettl3), which is responsible for adding methyl groups to RNA; demethylases-m6A ‘erasers’, such as fat mass and obesity-associated protein (FTO) and a-ketoglutarate-dependent dioxygenase alkB homolog 5 (Alkbh5), which delete methyl groups and remove m6A modifications; and m6A ‘readers’, such as YTH domain-containing family protein 1 (Ythdf1) and YTH domain-containing family protein 2 (Ythdf2), which identify downstream RNAs and regulate their metabolism. The results revealed a significant increase in the mRNA level of FTO (*p* = 0.0311) in the mPFC of the HFD group (Figure 6A). Subsequently, we examined the mRNA levels of MMP9 in the mPFC, which were dramatically elevated in the HFD group (*p* = 0.0194) (Figure 6B).

## 4. Discussion

Anxiety drives individuals to consume more comfort food, and the intake of comfort food can lead to a sense of pleasure. However, consuming excessive amounts of comfort food can lead to obesity and metabolic disorders. Our study was designed to explore the impact and underlying mechanisms of an HFD on anxiety behaviors in mice that have been on an HFD since the age of 5 weeks. The findings indicated a notable decrease in anxiety levels in mice that were subjected to this dietary regime, consistent with the findings reported by Maniam and Morris and Finger showing that HFD can relieve anxiety [6,34]. Contrary to our findings, some studies have reported that adult mice fed with a long-term HFD exhibit more anxiety-like, depression-like, and disruptive social behaviors [35,36]. The most significant difference between our research protocol and the protocols of studies reporting anxiety is the duration of the HFD. In our study, the HFD period is four weeks, similar to the relatively short HFD period in the study by Maniam and Morris [34]. On the contrary, studies reporting that HFD leads to anxiety typically involve long-term HFD. For instance, male mice subjected to a 9-week regimen on a high-fat/high-sugar cafeteria diet did not exhibit heightened anxiety-like behaviors in the EPM. However, such behavioral changes became apparent following an extended period of 18 weeks of dietary exposure [18,37]. Indeed, our earlier research found that a four-week HFD could have anxiolytic effects, whereas a prolonged HFD did not. These findings suggest that the impact of an HFD on the central nervous system may vary depending on the duration of the exposure period. Further investigations are required for elucidating the mechanisms whereby a four-week HFD elicits anxiolytic-like behaviors in adolescent mice.

In the present study, we found that the BDNF-TrkB signaling pathway in the mPFC plays a crucial role in the regulation of anxiety-like behaviors. We noted an increase in the protein levels of mBDNF and P-TrkB in the mPFC of the HFD group (see Figure 2). These alterations in protein expression coincided with a decrease in anxiety levels among the subjects consuming the HFD, accompanied by an increase in dendritic spine density and the augmentation of postsynaptic proteins. The elevation of BDNF may be a result of stress or could represent a protective response. The anxiolytic effects of the HFD were abolished by using BDNF-neutralizing antibodies (see Figure 3), suggesting that the increase in BDNF is a protective response. Our results indicate that BDNF-TrkB signaling in the mPFC plays a role in facilitating the anxiolytic-like behaviors triggered by a four-week HFD.

Regarding the effects of an HFD on BDNF, the literature reports both increases and decreases in BDNF levels in the prefrontal cortex. For example, Virtuoso et al. reported that a chronic cafeteria diet on a restricted schedule increases BDNF levels in the prefrontal cortex (PFC) of rats [38]. Nevertheless, Miao et al. found that in pregnant female mice, HFD induced decreases in BDNF in the hippocampus and prefrontal cortex, but increases in the amygdala [39]. The divergent outcomes of HFD’s impact on BDNF may be attributable to variations in the duration of the HFD and the animal models used, including distinctions between adolescent and adult mice. One of the limitations of our study is that we did not conduct experiments on adult mice. To confirm that the increase in BDNF in the mPFC occurs only in mice that begin an HFD in adolescence, a comparison should be made with mice that start an HFD in adulthood. 

A four-week HFD resulted in increased levels of mBDNF rather than an increase in proBDNF (Figure 2). mBDNF is associated with neuronal survival, growth, and plasticity, whereas proBDNF is implicated in facilitating long-term depression (LTD), spine shrinkage, and apoptosis [40]. Indeed, in the present study, the PSD95 and dendritic spine density increased in the HFD group (Figure 5). Our findings indicate that the increase in mBDNF protein may be attributable to enhanced hydrolysis of proBDNF. Precursor proBDNF is synthesized in the endoplasmic reticulum and then undergoes a maturation process involving proteolytic cleavage to form the mature BDNF [41]. Notably, knockout of FTO in mice has been reported to result in defective synthesis of mature BDNF, which is attributed to decreased expression of MMP9 [33]. Our findings demonstrate that the mRNA levels of MMP9 and FTO were increased in the HFD group (Figure 6), suggesting the potential involvement of FTO in the processing of mBDNF via MMP9.

## 5. Conclusions

Our study investigated the effects of a four-week HFD on anxiety-like behaviors in mice, and the underlying molecular mechanisms. We observed that mice subjected to the HFD exhibited reduced anxiety levels, which correlated with alterations in the BDNF-TrkB signaling pathway in the mPFC. Additionally, we demonstrated that the anxiolytic effects of the HFD were attenuated by neutralizing antibodies against BDNF, implicating the involvement of the BDNF-TrkB pathway in mediating these behaviors. Our findings suggest a potential role for enhanced hydrolysis of proBDNF, possibly involving MMP9 and FTO, in the observed increase in mBDNF expression. Further research is needed to fully elucidate these mechanisms and explore potential therapeutic avenues for anxiety disorders associated with dietary patterns.

## Figures and Tables

**Figure 1 brainsci-14-00389-f001:**
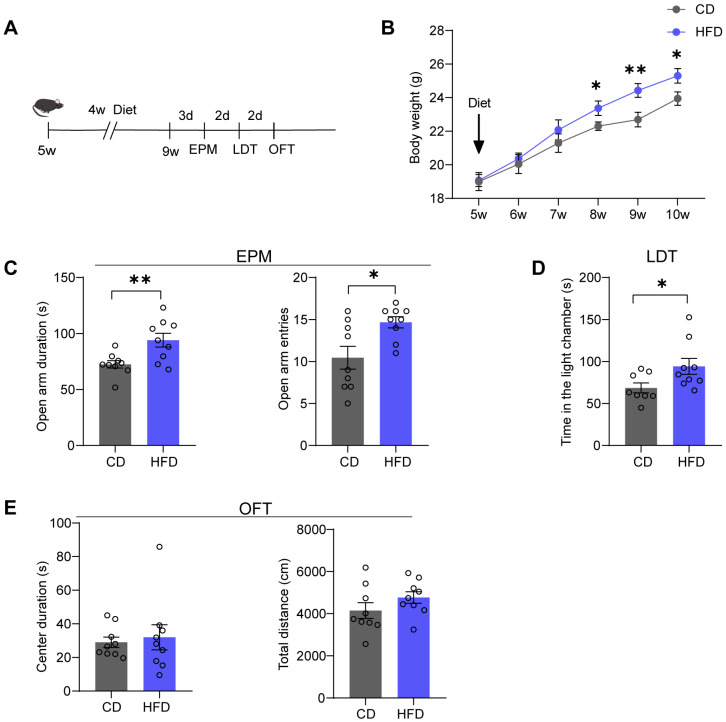
Mice fed an HFD for four weeks exhibit anxiolytic-like behaviors. (**A**) Schematic of diet processing and behavioral test paradigms. (**B**) Cumulative body weight curves of mice fed a standard chow diet (CD) and high-fat diet (HFD) (two-way repeated measures ANOVA followed by Bonferroni’s multiple comparisons test, time: F(1, 20) = 5.095, *p* = 0.0353; weight: F(5, 100) = 58.08, *p* < 0.0001; n = 11 mice per group). (**C**) Time spent exploring the open arms (**left**), entries into the two open arms (**right**) in the elevated plus maze test (EPM) (two-tailed Student’s *t*-test, duration: *t* = 3.071, *p* = 0.0073; entries: *t* = 2.796, *p* = 0.0130; n = nine mice per group). (**D**) Time spent in the light chamber in the light-dark box test (LDT) (two-tailed Student’s *t*-test, *t* = 2.210, *p* = 0.0430; CD: n = eight mice; HFD: n = nine mice). (**E**) Time spent in the center zone (center duration) during the first 5 min (**left**) and total distance explored during the 30 min (**right**) in the open field test (OFT) (two-tailed Student’s *t*-test, duration: *t* = 0.3616, *p* = 0.7224; distance: *t* = 1.341, *p* = 0.1987; n = nine mice per group). All data are presented as the mean ± SEM. * *p* < 0.05; ** *p* < 0.01.

**Figure 2 brainsci-14-00389-f002:**
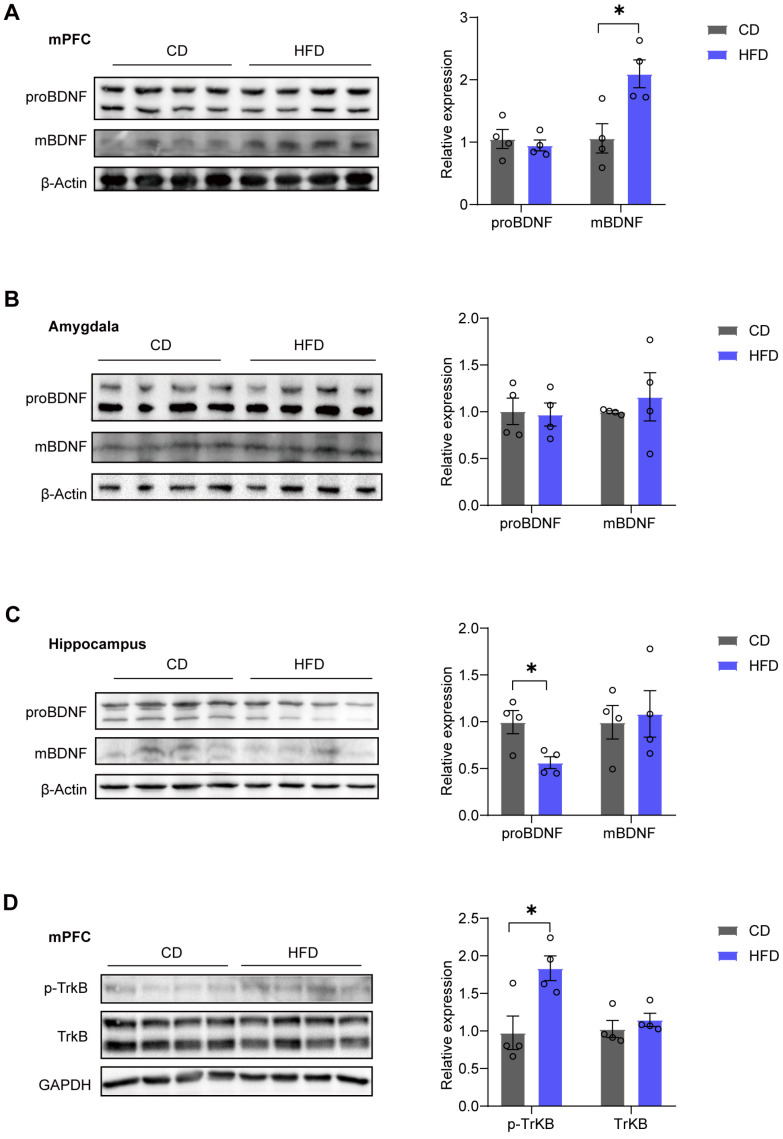
mBDNF and p-TrkB levels in the mPFC are increased in HFD mice. (**A**) Western blotting images (**left**) and quantification (**right**) of pro-protein of brain-derived neurotrophic factor (proBDNF) and mature brain-derived neurotrophic factor (mBDNF) in the medial prefrontal cortex (mPFC) (two-tailed Student’s *t*-test; proBDNF: *t* = 0.5867, *p* = 0.5788; mBDNF: *t* = 3.203, *p* = 0.0185; n = four mice per group). (**B**) Western blotting images (**left**) and quantification (**right**) of proBDNF and mBDNF in the amygdala (two-tailed Student’s *t*-test, proBDNF: *t* = 0.1815, *p* = 0.8619; mBDNF: *t* = 0.6283, *p* = 0.5530; n = four mice per group). (**C**) Western blotting images (**left**) and quantification (**right**) of proBDNF and mBDNF in the hippocampus (two-tailed Student’s *t*-test, proBDNF: *t* = 3.104, *p* = 0.0210; mBDNF: *t* = 0.2934, *p* = 0.7791; n = four mice per group). (**D**) Western blotting image (**left**) and quantification (**right**) of phosphorylated tyrosine kinase receptor B (p-TrkB) and tyrosine kinase receptor B (TrkB) in the mPFC (two-tailed Student’s *t*-test, p-TrkB: *t* = 3.104, *p* = 0.0210; TrkB: *t* = 0.8412, *p* = 0.4325; n = four mice per group). All data are presented as the mean ± SEM. * *p* < 0.05.

**Figure 3 brainsci-14-00389-f003:**
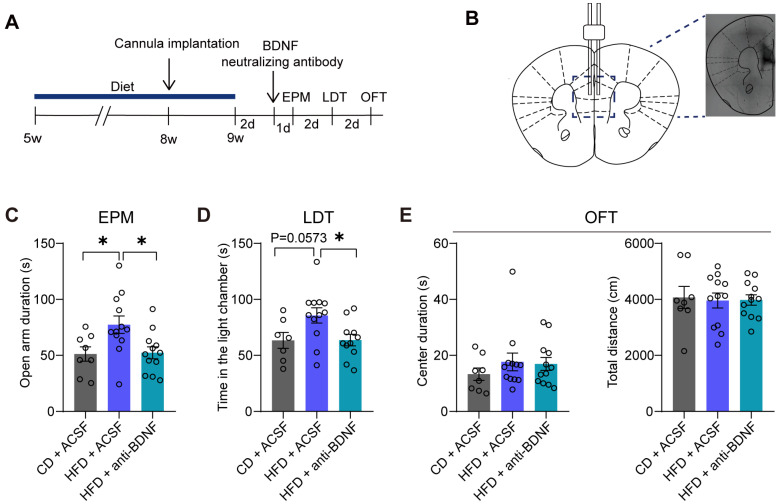
The administration of a BDNF-neutralizing antibody into the mPFC reverses anxiolytic-like behaviors induced by the HFD. (**A**) Schematic of surgical processing and behavioral test paradigms. (**B**) Representative images of the injection site in the medial prefrontal cortex (mPFC) of trypan blue. (**C**) Time spent exploring the open arms in the elevated plus maze test (EPM) (one-way ANOVA, F(2, 29) = 5.060, *p* = 0.0130; CD + ACSF: n = eight mice; HFD + ACSF: n = 12 mice; HFD + anti-BDNF: n = 12 mice). (**D**) Time spent in the light chamber in the light-dark box test (LDT) (one-way ANOVA, F(2, 27) = 4.292, *p* = 0.0241; CD + ACSF: n = seven mice; HFD + ACSF: n = 12 mice; HFD + anti-BDNF: n = 11 mice). (**E**) Time spent in the center zone (center duration) during the first 5 min (one-way ANOVA, F(2, 29) = 0.6306, *p* = 0.5394; CD + ACSF: n = eight mice; HFD + ACSF: n = 12 mice; HFD + anti-BDNF: n = 12 mice) (**left**) and total distance explored during the 30 min (**right**) in the open field test (OFT) (one-way ANOVA, F(2, 29) = 0.03793, *p* = 0.9628; CD + ACSF: n = eight mice; HFD + ACSF: n = 12 mice; HFD + anti-BDNF: n = 12 mice). All data are presented as the mean ± SEM. * *p* < 0.05.

**Figure 4 brainsci-14-00389-f004:**
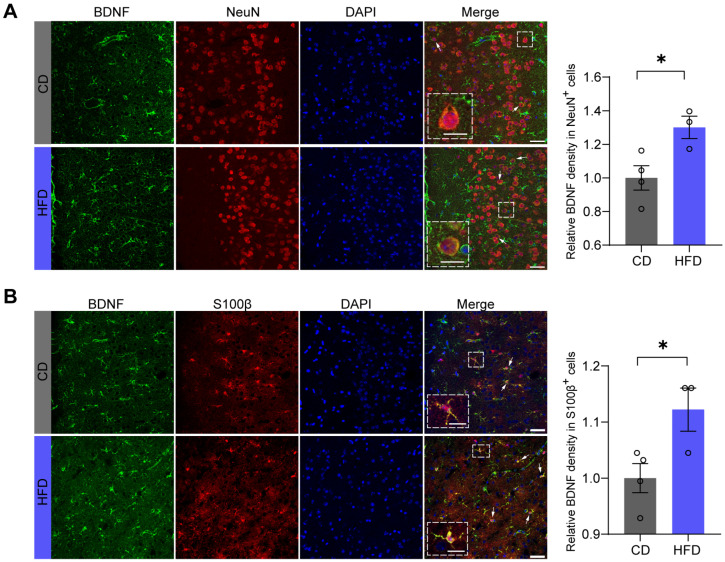
BDNF levels are increased in both neurons and astrocytes in the mPFC of mice fed an HFD. (**A**) Representative immunofluorescence staining (**left**) and quantification (**right**) of brain-derived neurotrophic factor (BDNF) (green) in NeuN^+^ cells (red) in the medial prefrontal cortex (mPFC) (two-tailed Student’s *t*-test, *t* = 2.939, *p* = 0.0323; CD: n = four mice; HFD: n = three mice; 80–100 cells per mice were observed). Scale bar = 10 μm (**left**), 25 μm (**right**). (**B**) Representative immunofluorescence staining (**left**) and quantification (**right**) of BDNF (green) in S100β^+^ cells (red) in the mPFC (two-tailed Student’s *t*-test, *t* = 2.731, *p* = 0.0412; CD: n = four mice; HFD: n = three mice; 30–50 cells per mice were observed). Scale bar = 10 μm (**left**), 25 μm (**right**). The arrows indicate co-labeled cells. All data are presented as the mean ± SEM. * *p* < 0.05.

**Figure 5 brainsci-14-00389-f005:**
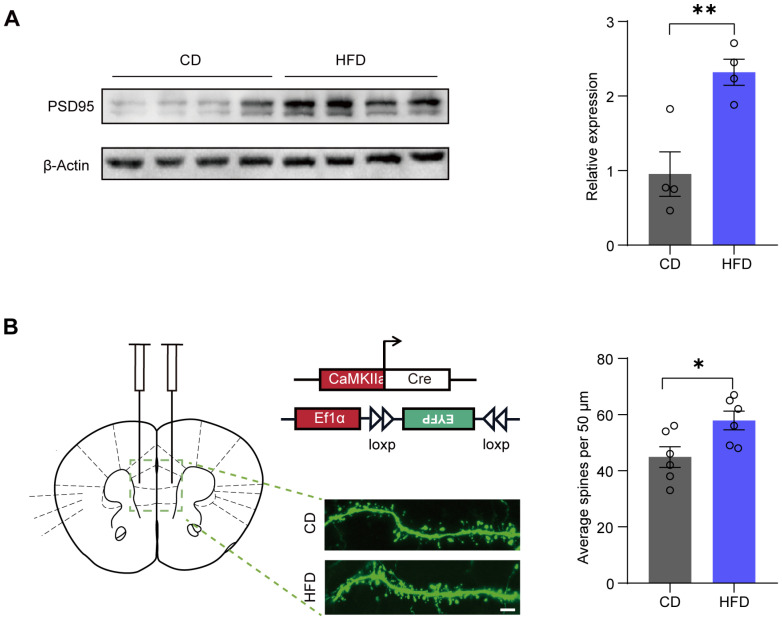
A four-week HFD promotes the expression of PSD95 and increases the density of dendritic spines. (**A**) Western blotting images (**left**) and quantification (**right**) of postsynaptic density protein 95 (PSD95) in the medial prefrontal cortex (mPFC) (two-tailed Student’s *t*-test, *t* = 3.935, *p* = 0.007; n = four mice per group). (**B**) Representative confocal imaging (**left**) and quantification (**right**) of dendritic spines of pyramidal neurons in the mPFC (two-tailed Student’s *t*-test, *t* = 2.625, *p* = 0.0254; n = six cells for three mice). Scale bar = 5 µm. All data are presented as the mean ± SEM. * *p* < 0.05; ** *p* < 0.01.

**Figure 6 brainsci-14-00389-f006:**
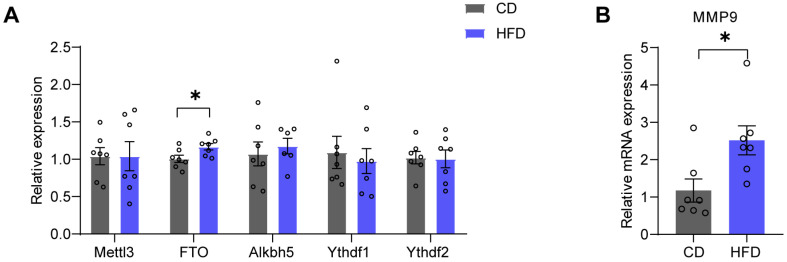
FTO and MMP9 levels in the mPFC are increased in HFD mice. (**A**) Relative mRNA levels of methyltransferase-like 3 (Mettl3), fat mass and obesity-associated protein (FTO), a-ketoglutarate-dependent dioxygenase alkB homolog 5 (Alkbh5), YTH domain-containing family protein 1 (Ythdf1), and YTH domain-containing family protein 2 (Ythdf2) in the mPFC (two-tailed Student’s *t*-test, Mettl3: *t* = 0.004694, *p* = 0.9963; FTO: *t* = 2.440, *p* = 0.0311; Alkbh5: *t* = 1.147, *p* = 0.2738; Ythdf1: *t* = 0.4234, *p* = 0.6795; Ythdf2: *t* = 0.1209, *p* = 0.9057; n = seven mice per group). (**B**) Relative levels of matrix metalloproteinase-9 (MMP9) mRNA in the mPFC (two-tailed Student’s *t*-test, *t* = 2.698, *p* = 0.0194; n = seven mice per group). All data are presented as the mean ± SEM. * *p* < 0.05.

**Table 1 brainsci-14-00389-t001:** Primer sequences for real-time PCR.

Genes		Primers (5′–3′)
*Mettl3*	F	GGACACGTGGAGCTCTATCC
	R	GCACGGGACTATCACTACGG
*FTO*	F	TCAGCAGTGGCAGCTGAAAT
	R	TCACCACGTCCCGAAACAAG
*Alkbh5*	F	CGGGAGAAGCTCAAGTCCAT
	R	TATTTCCGCTTGGTGGTCCC
*Ythdf1*	F	GTGGACCCCCAGAGAACAAA
	R	CCTCGCTGAGGGAGTAAGGA
*Ythdf2*	F	CCACAGGCAAGGCCGAATAA
	R	GACCAAGCAGCTTCACCCAA
*MMP9*	F	GCAGAGGCATACTTGTACCG
	R	TGATGTTATGATGGTCCCACTTG
*GAPDH*	F	TCAAGAAGGTGGTGAAGCAG
	R	TGGGAGTTGCTGTTGAAGTC

## Data Availability

All analyses and law data generated in this study are reliable and can be provided, upon reasonable request, by the authors. The data are not publicly available for privacy reasons.

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
