# Peer review of "A Four-Week High-Fat Diet Induces Anxiolytic-like Behaviors through Mature BDNF in the mPFC of Mice"

_brainsci, 2024, doi:10.3390/brainsci14040389_

Round 1
Reviewer 1 Report
Comments and Suggestions for Authors
The article by Huixian Huang et al. presents a comprehensive study on the effects of a high-fat diet on anxiolytic-like behaviors in mice. Additionally, cannula implantation, virus infusion, behavioral tests, and biochemical assays were carried out. The results highlight that a four-week high-fat diet induced anxiolytic-like behaviors and increased levels of mBDNF and p-TrkB in the mPFC. Furthermore, outcomes obtained in this research suggested that a high-fat diet may induce FTO and MMP9, which could potentially regulate BDNF processing, contributing to anxiolytic-like behaviors. The results are noteworthy; however, in my opinion, the manuscript in its current form requires thorough corrections.
1. The abstract is not informative enough.
2. The purpose of the research is not well defined.
3. The aim and scope of the study should be provided in the last paragraph of the Introduction and should be derived from information provided in the previous parts.
4. The introduction should not contain information about the results of the presented research and conclusions.
5. Authors are inconsequent in the use of abbreviations. At first use, the abbreviation should be explained and then used consistently throughout the manuscript.
6. Please provide the total number of mice used in the study.
7. Was each animal used only once in the study?
8. Please provide the manufacturer (company name, city, country, catalog number (if applicable)) of all substances/equipment/software used in experiments.
9. Why were only male mice used in studies? Gender difference is known to impart differences in behavioral patterns in humans as well as in rodents. Please address this point in the manuscript.
10. Please provide relevant references to the methodology, e.g. OFT, LDT, etc.
11. The results should be separated into individual tests/examinations.
12. The Results section should only contain outcomes. Please complete your results with statistical data.
13. The first sentence in each results subsection is unnecessary.
14. The results section should not include summaries/suggestions/comparisons with results obtained by other research teams.
15. Clearly describe the study groups.
16. Figure 1 A, 3 A should be moved to the Materials and Methods section.
17. Figure 1 should be divided into individual behavioral tests to increase readability.
18. The discussion is insufficient and repeats much information from the introduction.
19. The article lacks a well-formed conclusion summarizing the study's findings and implications.

Author Response
Dear Reviewer,
Thank you for your positive and constructive comments and suggestions. We have substantially revised our manuscript after reading the comments provided by the reviewers and uploaded the revised version together with our responses to the reviewers. In the revised manuscript, all the changes are highlighted in red for easy inspection. We hope these revisions can make our paper more acceptable. The revisions were addressed point by point below.
2. Point-by-point response to Comments and Suggestions for Authors
Comments 1: The abstract is not informative enough.
Response 1: Thank you for your valuable comment. We have revised the abstract to provide more information as suggested (line13-28). Specific changes have been highlighted in red in the revised manuscript for your review.
Comments 2: The purpose of the research is not well defined.
Response 2: Thank you for your comment. we have revised the final paragraph of the introduction (line 72-81) in the revised manuscript to explicitly define our research objectives as suggested. We have provided a more concise and focused statement to ensure that the purpose of our study is clearly communicated to the readers as follows:
In the present study, we investigated the impact of a four-week HFD on anxiolytic-like behaviors in mice at five weeks of age, correlating with the adolescent phase in human development. We aimed to elucidate the relationship between HFD consumption and anxiety, with a focus on the role of BDNF in the medial prefrontal cortex (mPFC). By examining changes in BDNF levels and the expression of N6-methyladenosine (m6A)-associated genes in the mPFC, we sought to provide insights into how HFD-induced alterations in BDNF processing may contribute to anxiolytic-like behaviors. Additionally, we aim to propose potential therapeutic targets for the treatment of anxiety disorders related to dietary patterns, particularly HFD consumption.
Your constructive feedback has been invaluable in improving the quality and clarity of our manuscript.
Comments 3: The aim and scope of the study should be provided in the last paragraph of the Introduction and should be derived from information provided in the previous parts.
Response 3: Thank you for your suggestion. The aim and scope of our study have been provided in the final paragraph of our Introduction section (line 72-81). The aim of our study is to investigate the impact of an HFD on anxiolytic-like behaviors in mice and to explore the underlying molecular mechanisms involved. We aim to elucidate the relationship between HFD consumption and anxiety, focusing on the role of BDNF in the medial prefrontal cortex (mPFC). By examining changes in BDNF levels and the expression of associated gene such as postsynaptic density protein 95 (PSD95), fat mass and obesity-associated protein (FTO), and matrix metalloproteinase-9 (MMP9) in the mPFC, we seek to provide insights into how HFD-induced alterations in BDNF processing may contribute to anxiolytic-like behaviors. Additionally, we aim to propose potential therapeutic targets for the treatment of anxiety disorders related to dietary patterns, particularly HFD consumption.
Comments 4: The introduction should not contain information about the results of the presented research and conclusions.
Response 4: Thank you for your suggestion. We have thoroughly revised introduction section of the manuscript (line 32-81) and can confirm that it does not contain any information regarding the results or conclusions of the presented research. We have made sure to adhere strictly to your feedback and focus solely on providing background information, outlining the significance of the research topic, and clearly stating the research aim and scope.
Comments 5: Authors are inconsequent in the use of abbreviations. At first use, the abbreviation should be explained and then used consistently throughout the manuscript.
Response 5: Thank you for your suggestion. We appreciate your attention to detail and the consistency in the use of abbreviations throughout the manuscript. We have made the necessary corrections in the revised manuscript. All abbreviations are now explained at their first use and used consistently thereafter to enhance clarity and readability. Reviewer’s comment has been instrumental in improving the quality of our manuscript.
Comments 6: Please provide the total number of mice used in the study.
Response 6: Thank you for your comment. We used a total of 80 mice throughout the entire experiment. After behavioral experiments with mice, tissues were collected for biochemical analyses such as Western blotting, qPCR, and immunofluorescence. We aim to minimize the number of mice used in experiments while achieving our research objectives. The specific number of mice used in each experiment is detailed in the figure legends. If you need any further clarification or information, please feel free to let us know.
Comments 7: Was each animal used only once in the study?
Response 7: Thank you for your comment. No, Some mice underwent multiple experiments. For instance, after completing behavioral experiments with mice (Figure 1), brain tissues were collected for biochemical analyses such as Western blotting, qPCR, and immunofluorescence. We aim to minimize the number of mice used in experiments while achieving our research objectives.
Comments 8: Please provide the manufacturer (company name, city, country, catalog number (if applicable)) of all substances/equipment/software used in experiments.
Response 8: Thank you for your valuable suggestion. We appreciate your attention to detail regarding the documentation of substances, equipment, and software used in our experiments. In the revised manuscript, we have endeavored to provided the manufacturer's information, including company name, city, country, and catalog number (if applicable), for all substances, equipment, and software used in the Materials and Methods section (line 82-236). Your suggestion has been instrumental in enhancing the completeness and transparency of our research.
Comments 9: Why were only male mice used in studies? Gender difference is known to impart differences in behavioral patterns in humans as well as in rodents. Please address this point in the manuscript.
Response 9: Thank you for your valuable suggestion. In this study, we only used male mice. This approach was adopted to minimize individual variations, and by employing males instead of females, we avoided cyclic hormonal effects linked to estrus. As you mentioned, gender can influence behavioral outcomes in both animals and humans. However, the purpose of our study did not involve investigating the impact of gender on behavioral outcomes.
Comments 10: Please provide relevant references to the methodology, e.g. OFT, LDT, etc.
Response 10: Thank you for your valuable suggestion. In the revised manuscript, we have provided relevant references to the methodology in the Materials and Methods section, such as OFT (references [22-25]), LDT (references [26, 27]), EPM (references [23, 28]), and sparse labeling of pyramidal neurons (references [29, 30]).
Comments 11: The results should be separated into individual tests/examinations.
Response 11: Thank you for your valuable suggestion. We appreciate your suggestion to separate the results into individual tests/examinations. While we understand the benefits of this approach, we are concerned that it may lead to a proliferation of figures and results, potentially compromising clarity and readability.
Comments 12: The Results section should only contain outcomes. Please complete your results with statistical data.
Response 12: Thank you for your valuable suggestion. In the revised manuscript, we have complete the results along with statistical data (p value) in the Results section (Paragraph 3.1, 3.2, 3.3, 3.4, 3.5, 3.6) as suggested.
Comments 13: The first sentence in each results subsection is unnecessary.
Response 13: Thank you for your valuable suggestion. In the revised manuscript, we have removed the first sentence of subsections 3.1, 3.2, and 3.3 in the Results section as suggested. Additionally, to improve clarity regarding the purpose of each step of our research, we have retained the first sentence of subsections 3.4, 3.5, and 3.6.
Comments 14: The results section should not include summaries/suggestions/comparisons with results obtained by other research teams.
Response 14: Thank you for your valuable suggestion. In the revised manuscript, we have removed the summaries from the Results section as suggested.
Comments 15: Clearly describe the study groups.
Response 15: Thank you for your valuable suggestion. The experimental groups were assigned randomly. We also specified in subsection 2.1 of the Materials and Methods section (line 89-90) that our experiments involved random group assignments.
Comments 16: Figure 1 A, 3 A should be moved to the Materials and Methods section.
Response 16: Thank you for your suggestion regarding Figure 1A and 3A. We appreciate your attention to detail and clear presentation in the manuscript. Clear and straightforward visuals, such as those presented in Figures 1A and 3A, can help people better grasp how our experiments were conducted. By including these figures as images, readers can get a clearer picture of our methods while going through the results. Therefore, it is beneficial to include these figures in the manuscript as images. Thank you once again for your valuable comment.
Comments 17: Figure 1 should be divided into individual behavioral tests to increase readability.
Response 17: Thank you for your valuable suggestion. To enhance readability, we have consolidated individual behavioral tests (Figure 1) into Figure 1C and Figure 1E, clearly labeling them as EPM and OFT in the figure images.
Comments 18: The discussion is insufficient and repeats much information from the introduction.
Response 18: Thank you for your suggestion. We have revised the discussion section to enhance its depth and avoid redundant information from the introduction. We value your contribution to improving the quality of our manuscript.
Comments 19: The article lacks a well-formed conclusion summarizing the study's findings and implications.
Response 19: Thank you for your suggestion. We have addressed this concern by adding a well-formed conclusion at the end of the manuscript (line 445-456). This conclusion summarizes the key findings of the study and discusses their implications. Here is the specific content we have added:
Our study investigated the effects of a four-week HFD on anxiety-like behaviors and the underlying molecular mechanisms in mice. We observed that mice subjected to the HFD exhibited reduced anxiety levels, which correlated with alterations in the BDNF-TrkB signaling pathway in the mPFC. Additionally, we demonstrated that the anxiolytic effects of the HFD were attenuated by neutralizing antibodies against BDNF, implicating the involvement of the BDNF-TrkB pathway in mediating these behaviors. Our findings suggest a potential role for enhanced hydrolysis of proBDNF, possibly involving MMP9 and FTO, in the observed increase in mBDNF expression. Further research is needed to fully elucidate these mechanisms and explore potential therapeutic avenues for anxiety disorders associated with dietary patterns.
Once again, we would like to express our gratitude to you for your valuable comments and guidance throughout the review process. Your contributions have been instrumental in strengthening the scientific rigor and integrity of our research.
Kind regards!

Reviewer 2 Report
Comments and Suggestions for Authors
In this manuscript, Huang and colleagues study the effect of HFD on anxiety behavior and found that mice exposed to HFD for 4 weeks exhibited anxiolytic-like behaviors. The authors demonstrate the anxiolytic effects of HFD due to the increased expression of BDNF and p-TrkB in the mPFC, while the administration of a BDNF neutralizing antibody blunts these anxiolytic effects. These findings suggest that the BDNF-TrkB signaling pathway in the mPFC mediates the subjects' anxiolytic behaviors.
This study provides a clear and comprehensive exploration of how HFD mediates anxiolytic behaviors. The results are straightforward and lend support to the conclusions. However, I have two major concerns that, if addressed, would enhance the manuscript.:
1. In the Results section, the authors conducted a series of behavioral tests to address this topic. However, the total number of mice used in this experiment appears to vary across these tests. Did the authors exclude any outliers from the behavioral tests? If so, the authors should define the criteria for identifying outliers and describe this process in the Statistical Analysis section.
2. In the Results section, the authors conducted a comprehensive analysis of anxiety-related behaviors. However, in Section 3.3, it is unclear how the BDNF neutralizing antibody affects the number of transitions between the open arms of the maze. This is significant because the Elevated Plus Maze (EPM) assay is a crucial measure of anxiety.
Author Response
Dear Reviewer,
Thank you for your positive and constructive comments and suggestions. We have substantially revised our manuscript after reading the comments provided by the reviewers and uploaded the revised version together with our responses to the reviewers. In the revised manuscript, all the changes are highlighted in red for easy inspection. We hope these revisions can make our paper more acceptable. The revisions were addressed point by point below.
Comments 1: In the Results section, the authors conducted a series of behavioral tests to address this topic. However, the total number of mice used in this experiment appears to vary across these tests. Did the authors exclude any outliers from the behavioral tests? If so, the authors should define the criteria for identifying outliers and describe this process in the Statistical Analysis section.
Response 1: Thank you for your valuable comment. In Figure 1C, the EPM experiment involved 9 mice per group. In Figure 1D, the LDT experiment had 8 mice in the CD group and 9 mice in the HFD group. In Figure 1E, the OFT experiment included 11 mice in the CD group and 12 in the HFD group. The inconsistency in the total number of mice used in these experiments arose because during the LDT experiment (Figure 1D, Figure 3D), there was a systematic error with the LD box, rendering the data unanalyzable and leading to the exclusion of individual mice from the experiment. Although the mice used in the OFT experiment (Figure 1E) were from a different batch than those used in Figures 1C and 1D, consistent results were obtained from repeated experiments, supporting the conclusions of our study. To prevent any misunderstanding by the reviewers and readers, we have replaced the mice used in the behavioral experiments in Figure 1 with a single batch. The specific number of mice used in each experiment is detailed in the figure legends. These changes do not affect the framework and outcomes of the manuscript.
Comments 2: In the Results section, the authors conducted a comprehensive analysis of anxiety-related behaviors. However, in Section 3.3, it is unclear how the BDNF neutralizing antibody affects the number of transitions between the open arms of the maze. This is significant because the Elevated Plus Maze (EPM) assay is a crucial measure of anxiety.
Response 2: Thank you for your valuable suggestion. We have provided the images of open arm entries about how the BDNF neutralizing antibody affects the number of transitions between the open arms of the maze. According to the images below for the response, we found that BDNF neutralizing antibody can significantly reduce the number of transitions between the open arms induced by the four-week high-fat diet.

Once again, we would like to express our gratitude to you for your valuable comments and guidance throughout the review process. Your contributions have been instrumental in strengthening the scientific rigor and integrity of our research.
Kind regards!

Reviewer 3 Report
Comments and Suggestions for Authors
In general, I think the subject of article by Huang and co-authors is really interesting, and the authors’ fascinating observations on this timely topic may be of interest to the readers of “Brain sciences”. However, some minor comments needed to be addressed:
1) Throughout the text, the authors refer to the immature form of BDNF as imBDNF, although based on the molecular weight (35 kDa) provided in the supplementary file, it is pre-pro-BDNF. I think that it is more correct to call this isoform as pre-pro-BDNF, since proBDNF with a mass of 25 kDa is also immature, but the authors did not study it. Since each BDNF isoform and even cleaved pro-peptides have their own functional activity, it is better to avoid terminological confusion in this subject.
2) Authors should justify in "introduction" the choice of the four week age of mice to start introducing the HFD. Also it is not clear, why on the Figures 1A and 3A the start age is indicated as 5 weeks.
3) The design of plasmids and assembly of virus vectors should be indicated at the “materials and methods” section.
Author Response
Dear Reviewer,
Thank you for your positive and constructive comments and suggestions. We have substantially revised our manuscript after reading the comments provided by the reviewers and uploaded the revised version together with our responses to the reviewers. In the revised manuscript, all the changes are highlighted in red for easy inspection. We hope these revisions can make our paper more acceptable. The revisions were addressed point by point below.
Comments 1: Throughout the text, the authors refer to the immature form of BDNF as imBDNF, although based on the molecular weight (35 kDa) provided in the supplementary file, it is pre-pro-BDNF. I think that it is more correct to call this isoform as pre-pro-BDNF, since proBDNF with a mass of 25 kDa is also immature, but the authors did not study it. Since each BDNF isoform and even cleaved pro-peptides have their own functional activity, it is better to avoid terminological confusion in this subject.
Response 1: Thank you for your valuable suggestion. We have reviewed relevant references, and as you mentioned, it is inappropriate to categorize pre-proBDNF and proBDNF as immature BDNF. To avoid any confusion, we have made the necessary changes in the manuscript. Following the report by Henry K et al.[1], we have made the necessary corrections in the revised manuscript. Specifically, we accurately labeled the 35 kDa band as proBDNF and provided supporting literature for this correction. Additionally, we have provided relevant references regarding the hydrolysis of proBDNF into mature BDNF[2], and the details explaining about pre-proBDNF and proBDNF[3]. The specific references are as follows:
[1] Teng, H. K. and K. K. Teng, et al. (2005). "ProBDNF induces neuronal apoptosis via activation of a receptor complex of p75NTR and sortilin." J Neurosci 25(22): 5455-63.
[2] Nagappan, G. and E. Zaitsev, et al. (2009). "Control of extracellular cleavage of ProBDNF by high frequency neuronal activity." Proc Natl Acad Sci U S A 106(4): 1267-72.
[3] Zanin, J. P. and N. Unsain, et al. (2017). "Growth factors and hormones pro-peptides: the unexpected adventures of the BDNF prodomain." J Neurochem 141(3): 330-340.
Comments 2: Authors should justify in "introduction" the choice of the four week age of mice to start introducing the HFD. Also it is not clear, why on the Figures 1A and 3A the start age is indicated as 5 weeks.
Response 2: Thank you for your valuable suggestion. We obtained 4-week-old male mice from the Guangdong Medical Laboratory Animal Center in China. Prior to the start of the experiment, it was necessary for the mice to acclimate to the new environment for one week. Therefore, the high-fat diet or standard chow diet feeding started at the age of five weeks and lasted for four weeks. The starting age of the mice is indicated in Figure 1A and Figure 3A. To avoid any misunderstandings, we have provided detailed explanations in the Animals subsection of the Materials and Methods section.
Comments 3: The design of plasmids and assembly of virus vectors should be indicated at the “materials and methods” section.
Response 3: Thank you for your valuable suggestion. The viral vector utilized in Figure 5B was obtained from BrainVTA Biotech Co., Ltd (Wuhan, China). This viral vector is a well-established commercial product. Information regarding the viral vector, including its design details, has been provided in the Stereotaxic Injection of Virus subsection of the Materials and Methods section (line 203), along with relevant references (references [29, 30]). We have also provided the manufacturer (BrainVTA, Wuhan, China) of the virus in the Stereotaxic Injection of Virus subsection of the Materials and Methods section. We hope this adequately addresses your query regarding the design information of the viral vector.
Once again, we would like to express our gratitude to you for your valuable comments and guidance throughout the review process. Your contributions have been instrumental in strengthening the scientific rigor and integrity of our research.
Kind regards!
Reviewer 4 Report
Comments and Suggestions for Authors
The work of Hung H. et al. is an animal experimental work that looks for molecular relationships between anxiolytic-like behaviors and a high-fat diet. The work brings several supporting results for the authors' conclusions. However, I have to say that I had difficulty finding my way around the job. In principle, it would be an enrichment to make some scheme of experiments, which would allow the reader to orientate himself (something similar to clinical studies, a Consort Flow diagram is presented, indicating the number of mice in which experiment). In the paper, he states that the mice received an HFD for 4 weeks, but Figure 1B implies that they received an HFD for 5 weeks.??
Another principled recommendation: In the text for each image, explain all abbreviations used - although most are given in the text)
- In fig. 3A - what do the authors mean by "surgery"? and Microinfusion means an infusion of BDNF?
- The paragraph (lines 146-158) should be placed before paragraph 2.3
- Paragraph 2.4. - more details about the dissected tissue
- It is not entirely clear from the text how many mice were in the HFD and CD groups in individual experiments (the recommended Diagram would contribute to this)
The work brings several significant results, but their presentation is sometimes less comprehensible. After editing, the work could significantly contribute to understanding the molecular basis of anxiety.
Author Response
Dear Reviewer,
Thank you for your positive and constructive comments and suggestions. We have substantially revised our manuscript after reading the comments provided by the reviewers and uploaded the revised version together with our responses to the reviewers. In the revised manuscript, all the changes are highlighted in red for easy inspection. We hope these revisions can make our paper more acceptable. The revisions were addressed point by point below.
Comments 1: In the paper, he states that the mice received an HFD for 4 weeks, but Figure 1B implies that they received an HFD for 5 weeks.
Response 1: Thank you for your valuable comment. Yes, the mice were exposed to a high-fat diet for 4 weeks. However, during the weight monitoring process depicted in Figure 1B, we continued to monitor the weight changes of mice on the high-fat diet for 5 weeks. Additionally, for our behavioral experiments and biochemical analyses, we utilized mice that had been on the high-fat diet for 4 weeks. To avoid any confusion, we have clarified in the Animals subsection of the Materials and Methods section in the revised manuscript (line 89-91) the age at which mice started the high-fat diet and the duration of the high-fat diet feeding.
Comments 2: Another principled recommendation: In the text for each image, explain all abbreviations used - although most are given in the text).
Response 2: Thank you for your valuable suggestion. In the revised manuscript, we have already explained all abbreviations in the each figure legend as suggested.
Comments 3: In fig. 3A - what do the authors mean by "surgery"? and Microinfusion means an infusion of BDNF?
Response 3: Thank you for your valuable comment. We appreciate your attention to detail and clear presentation in the manuscript.
"Surgery" (Figure 3A) refers to the procedure of implanting cannulas into the medial prefrontal cortex (mPFC) of the mice.
Microinfusion means an infusion of BDNF neutralizing antibody. After a recovery period of at least one week, microinfusion pumps were used to administer BDNF neutralizing antibodies into the mPFC through the implanted cannulas 24 hours prior to the behavioral experiments.
To avoid any confusion, we have already indicated "cannula implantation” and “BDNF neutralizing antibody" in Figure 3A.
Comments 4: The paragraph (lines 146-158) should be placed before paragraph 2.3.
Response 4: Thank you for your valuable suggestion. We have followed the reviewer's suggestion and moved Paragraph 2.5 (immunofluorescence) from its original placement after Paragraph 2.3 (Western blotting) to before it. Additionally, we have adjusted the numbering accordingly in the revised manuscript.
Comments 5: Paragraph 2.4. - more details about the dissected tissue.
Response 5: Thank you for your valuable suggestion. We have provided more detailed information about the dissected tissue in Paragraph 2.5 ( Paragraph 2.4 in the original manuscript) of the revised manuscript (line 161-164). Specifically, we describe the process of swiftly decapitating the mice, followed by extraction of the mouse brains. Subsequently, we dissected the medial prefrontal cortex (mpfc) of mice using a mouse brain mold (RWD, Shenzhen, China, 68708), based on coronal sections from bregma 2.4 mm to bregma 1.4 mm as indicated in the mouse brain atlas.
Comments 6: It is not entirely clear from the text how many mice were in the HFD and CD groups in individual experiments (the recommended Diagram would contribute to this).
Response 6: Thank you for your valuable comment. As suggested by the reviewer, we have made the number of mice used in each experimental group (CD and HFD) clearer. This information is now included in the caption of each figure in the revised manuscript.
Once again, we would like to express our gratitude to you for your valuable comments and guidance throughout the review process. Your contributions have been instrumental in strengthening the scientific rigor and integrity of our research.
Kind regards!
Round 2
Reviewer 1 Report
Comments and Suggestions for Authors
None
Reviewer 2 Report
Comments and Suggestions for Authors
Authors responded adequately to comments of this reviewer, and revised the manuscript accordingly. This reviewer appreciates authors' efforts to improve the manuscript.
Reviewer 4 Report
Comments and Suggestions for Authors
The authors accepted all the reviewer's comments. Thank you